## TOPICAL REVIEW

# The link between sarcopenic obesity and Alzheimer's disease: a brain-derived neurotrophic factor point of view

Emily N. Copeland[1,2], Paul J. LeBlanc[2,3] (iD), Paula Duarte-Guterman[4,5] (iD), Val A. Fajardo[1,3] (iD) and Rebecca E. K. MacPherson[2,5] (iD)

[1]*Department of Kinesiology, Brock University, St. Catharines, ON, Canada*
[2]*Department of Health Sciences, Brock University, St. Catharines, ON, Canada*
[3]*Centre for Bone and Muscle Health, Brock University, St. Catharines, ON, Canada*
[4]*Department of Psychology, Brock University, St. Catharines, ON, Canada*
[5]*Centre for Neurosciences, Brock University, St. Catharines, ON, Canada*

Handling Editors: Laura Bennet & Mathew Piasecki

The peer review history is available in the Supporting Information section of this article (https://doi.org/10.1113/JP288032#support-information-section).

**Abstract figure legend** *A*, Sarcopenic obesity is defined as the age-related loss of skeletal muscle mass and function that often leads to the progression of comorbidities, such as Alzheimer's disease (AD). Though the exact link between the two diseases is unknown, alterations in brain-derived neurotrophic factor (BDNF) may be a contributor. Lower BDNF concentrations are linked to a higher risk of AD, through the increased production of amyloid beta (A$\beta$) and neurofibrillary tangles (NFTs). Lower BDNF concentrations also impair mitochondrial function, increasing the likelihood of insulin resistance and obesity because BDNF concentration is inversely associated with obesogenic markers in adipose tissue. Additionally, lower BDNF is associated with impaired skeletal muscle fibre repair and regeneration. *B*, Alternatively, higher BDNF concentrations are associated with a lower risk of AD, reduced adiposity through an increase in thermogenesis and adipocyte browning and increased muscle mass. Together, the current literature suggests that BDNF may be a viable target for multiple age-related diseases, though more research is needed to conclude this idea, and special considerations should be made to evaluate the influence of biological sex, as women are at a higher risk of developing both AD and sarcopenic obesity.

The Journal of Physiology

**Abstract** Age-related diseases are becoming more prominent as the lifespan of the global population rises. Many of these diseases coincide with each other and can even influence the onset of additional comorbidities. Sarcopenic obesity is described as age-related loss of muscle mass that concurs with excessive weight gain and tends to increase the risk of comorbidity development, including Alzheimer's disease (AD). Though the exact link between sarcopenic obesity and AD is not known, this review explores the possibility that reduced levels of brain-derived neurotrophic factor (BDNF) throughout the body may serve as the underlying commonality. In AD, reductions in BDNF signalling through its receptor promote the activation of glycogen synthase kinase 3 beta (GSK3$\beta$), which subsequently increases the production of amyloid beta (A$\beta$) peptides and neurofibrillary tangles (NFTs). In the skeletal muscle, lower BDNF concentrations are linked to impaired muscle fibre repair and regeneration, increasing the likelihood of sarcopenia. Furthermore, the absence of BDNF impairs mitochondrial function, leading to insulin resistance and increased adiposity. BDNF concentration has a negative relationship with obesogenic markers in adipose tissue, and as such, lower concentrations of BDNF lead to weight gain. Collectively, current literature suggests that BDNF attenuates AD pathology while improving skeletal muscle mitochondrial function, whole-body insulin resistance and facilitating adipocyte browning. Therefore, BDNF may be a viable target for multiple age-related diseases, but more research is required to substantiate this claim, with a particular focus on examining any potential influence of biological sex, as women are at a higher risk for both AD and sarcopenic obesity.

(Received 31 October 2024; accepted after revision 16 January 2025; first published online 12 February 2025)

**Corresponding author** R. E. K. MacPherson: Associate Professor, Department of Health Sciences, Brock University, 1812 Sir Isaac Brock Way, St. Catharines, ON, L2S 3A1, Canada. Email: rmacpherson@brocku.ca

## Introduction

The lifespan of the global population is rising, resulting in an increasing number of elderly adults. According to the World Health Organization, by 2030, 1 out of every 6 individuals globally will be $\geq$ 60 years old (World Health Organization, 2024). By 2050, the global population of individuals aged 60 and above will double to 2.1 billion, with the number of people aged 80 and above reaching 426 million (World Health Organization, 2024). A significant concern for older adults is the coexistence of low muscle mass and increased body fat, a condition known as sarcopenic obesity. Alone, the loss of muscle mass (sarcopenia) or an excess of adiposity (obesity) can increase the risk of several age-related disorders, including type 2 diabetes, cardiovascular disease and Alzheimer's disease (AD). However, recent evidence demonstrates that the co-existence of sarcopenia and obesity in sarcopenic obesity synergistically increases the risk of developing these age-related conditions. In this review, we discuss how sarcopenic obesity can increase the risk of AD. Further, we will focus on the role of brain-derived neuro-trophic factor (BDNF) content and signalling in multiple tissues and the influence that alterations to BDNF might have on the onset of AD.

AD is the most common form of dementia, accounting for 60%–80% of all diagnoses. The prevalence of AD is steadily increasing and is expected to affect over 65 million people by 2030 and more than 131 billion people by 2050 (Dartigues, 2009; Wu et al., 2017). AD is considered to have two pathological hallmarks: neuro-fibrillary tangles (NFTs) and amyloid beta (A$\beta$) plaques. NFTs originate from tau, a microtubule-associated protein that aids in cellular communication via vesicular transport and is tightly regulated through its phosphorylation status (Barbier et al., 2019; Haque et al., 2004). During AD

**Emily N. Copeland** received her BSc in neuroscience and MSc in applied health sciences at Brock University. She is currently pursuing her PhD at Brock University under the supervision of Dr Val A. Fajardo and Dr Rebecca E.K. MacPherson. Her NSERC-funded doctoral studies focus on the influence of muscle health on brain health, specifically how muscle wasting (i.e. muscular dystrophy and spaceflight) increases the risk of Alzheimer's disease.

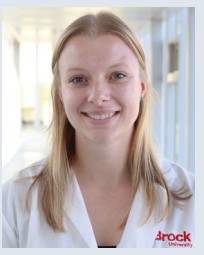

pathogenesis, tau becomes hyperphosphorylated, causing its aggregation and the formation of NFTs (Grundke-Iqbal et al., 1988; Iqbal et al., 1986, 1989). On the contrary, A$\beta$ plaques originate from a transmembrane protein known as amyloid precursor protein (APP). Briefly, APP is processed through either the amyloidogenic or non-amyloidogenic pathway. The amyloidogenic pathway leads to the production of an A$\beta$ fragment varying between 38 and 40 amino acids in length that can accumulate and aggregate, forming insoluble A$\beta$ plaques (Hampel et al., 2021).

AD can be further classified as familial or sporadic. Sporadic AD accounts for 95% of all AD diagnoses and is associated with non-modifiable risk factors, including age, sex and genetics (Bekris et al., 2010), and modifiable risk factors, including metabolic health and obesity (Edwards Iii et al., 2019), thus linking brain health to the health of peripheral tissues such as skeletal muscle and adipose tissue. In fact, AD has become known as a metabolic disease (Kang et al., 2017), and therefore improving whole-body metabolism may attenuate AD pathology. As we age, our bodies undergo progressive changes in body composition, shifting away from primarily lean muscle mass towards fat mass. The loss of lean mass is often associated with the reduction in size and strength of skeletal muscle, known as sarcopenia (Cruz-Jentoft et al., 2010, 2019), whereas an excessive gain of fat mass and insufficient energy expenditure often lead to obesity (Spiegelman & Flier, 2001). The loss of muscle mass and strength increases the risk of frailty and is associated with other comorbidities, including obesity and type 2 diabetes (Murawiak et al., 2022; Silveira et al., 2022; Steffl et al., 2017). Additionally, those with sarcopenia are six times more likely to have combined physical and cognitive impairments (Pacifico et al., 2020; Pacifico et al., 2022), and the prevalence of sarcopenia is significantly greater in AD patients compared to controls by up to 60% (Ogawa et al., 2018). Excess adiposity in obesity also increases the risk for type 2 diabetes (Wang et al., 2005), leading to a greater risk of AD. In fact, AD has become known as type 3 diabetes as peripheral metabolic syndrome has been linked to brain insulin resistance, ultimately resulting in neurodegeneration (Caberlotto et al., 2019; Nguyen, Ta, Nguyen, Le, et al., 2020; Nguyen, Ta, Nguyen, Nguyen, et al., 2020). Furthermore, mid-life obesity is associated with a 74% increased risk of dementia (Whitmer et al., 2005), and central obesity in the elderly is positively associated with AD development (Kalinkovich & Livshits, 2017). Thus, an ageing global population and the age-related synergies of both sarcopenia and obesity in sarcopenic obesity increasing the risk of AD present a significant health concern for the future.

Though the exact link between sarcopenic obesity and AD is not well understood, sarcopenic obesity is associated with hyperglycemia, hyperinsulinemia and a chronic inflammatory state, all of which are known to contribute to AD pathology (Kalinkovich & Livshits, 2017). BDNF, a neurotrophic factor, may provide a link between sarcopenic obesity and AD as it has been shown to be lower in patients with AD (Ng et al., 2019), sarcopenia (Miyazaki et al., 2021) and those living with obesity (Craft et al., 1993; Hong & Choi, 2020; Kinney et al., 2018; Luchsinger et al., 2004). Previous research has shown that BDNF directly impacts cognitive health and function in the context of AD, and these effects may be mediated through peripheral effects on muscle and adipose tissue, which will be the focus of this review.

## BDNF and AD pathology

First discovered in the brain in 1982, BDNF is one of the most abundant neurotrophic factors in the central nervous system (Barde et al., 1982). Synthesized as pre-pro-BDNF in the endoplasmic reticulum, the precursor signal is cleaved, allowing for transport of pro-BDNF to the Golgi where it is packaged into vesicles. The conversion of pro-to-mature BDNF is facilitated through endoproteases and/or proprotein convertases (Palomer et al., 2016), occurring in an activity-dependent manner (Je et al., 2012). Mature BDNF signals through the tropomyosin receptor kinase B (TrkB) receptor, inducing dimerization and the initiation of numerous pathways, including phosphoinositide 3-kinase and mitogen-activated protein kinase (Schirò et al., 2022). Alternatively, pro-BDNF loosely signals through the p75 neurotrophin receptor (NTR) to activate both survival and apoptotic pathways (Schirò et al., 2022). Both mature and pro-BDNFs are highly involved in neuronal development and survival, and synaptic plasticity in the adult brain. Mature BDNF signalling is associated with enhanced long-term potentiation (Ying et al., 2002), whereas pro-BDNF is linked to long-term depression (Woo et al., 2005), suggesting that a balance must be maintained to enhance learning and memory. Fluctuations in the pro-to-mature BDNF ratio are observed during normal ageing; however, low concentrations of mature BDNF increase the risk of AD, and it has been suggested that the downregulation of BDNF directly precedes the onset of AD (Ginsberg et al., 2019).

From a mechanistic lens, BDNF has been linked to the production of both A$\beta$ peptides and NFTs; however, the focus will be on A$\beta$. Previous work has shown that lower BDNF mRNA expression and lessened BDNF signalling is associated with increased soluble A$\beta$ in both rodent and human cortical tissue (Garzon et al., 2002; Tong et al., 2004). Additionally, transgenic AD mouse models that express high A$\beta$42/A$\beta$40 ratios show significant reductions in BDNF mRNA expression (Peng et al., 2009), suggesting that there is a relationship

between specific A$\beta$ oligomers and BDNF reductions; however the order which occurs first, A$\beta$ accumulation or BDNF reductions, is unknown. In humans, serum analyses of BDNF have shown significant lower BDNF concentrations in AD patients compared to cognitively healthy controls (Mori et al., 2021; Siuda et al., 2017), and a lower serum BDNF positively correlates with cerebrospinal fluid concentrations of A$\beta$42 (Mori et al., 2021). Further, BDNF has been shown to regulate $\beta$-site amyloid precursor protein cleaving enzyme-1 (BACE1) activity, the rate-limiting enzyme in amyloidogenic APP processing. Specifically, direct treatment of neuronal cells (SH-SY5Y) and prefrontal cortex explants from male C57BL/6J mice with BDNF resulted in lower BACE1 activity (Baranowski et al., 2021, 2024). Additionally, *in vivo*, peripheral injections of BDNF in male C57BL/6J mice resulted in lower BACE1 activity in the prefrontal cortex following 8 weeks of treatment (Baranowski et al., 2021). As BACE1 cleavage of APP is the rate-limiting step in A$\beta$ production, this body of work indicates that BDNF may contribute to the balance between the amyloidogenic and nonamyloidogenic pathway activity and therefore A$\beta$ accumulation.

BDNF signalling, through TrkB, can also inhibit glycogen synthase kinase 3 beta (GSK3$\beta$), a constitutively active serine-threonine kinase that contributes to APP cleavage through BACE1 (Aplin et al., 1996; Baranowski et al., 2024). GSK3$\beta$ is inhibited through the phosphorylation of its serine 9 site, preventing substrate binding and the subsequent activation of its kinase activity. Cells expressing the human Swedish mutant APP treated with a potent GSK3 inhibitor (AR-A014418) showed lower expression of APP C99, a BACE1 cleavage product (Ly et al., 2013). When APP23/PS45 double transgenic male and female mice were treated for 4 weeks with the same GSK3 inhibitor, brain BACE1 cleavage products and A$\beta$ content were significantly lower, whereas spatial memory was improved (analysis by sex was not performed) (Ly et al., 2013). On the contrary, BDNF may be an equally effective GSK3 inhibitor, as neuronal cells (SH-SY5Y) treated with BDNF showed higher phosphorylated GSK3$\beta$ protein (Baranowski et al., 2024). However, it is important to note that BDNF likely facilitates this GSK3$\beta$ inhibition through Akt as cells that were dually treated with BDNF and Wortmannin, an Akt inhibitor, had less GSK3$\beta$ phosphorylation (Baranowski et al., 2024). Further, BDNF treatment lowered BACE1 activity, while increasing alpha-secretase (ADAM10) activity, and therefore promoting the non-amyloidogenic metabolism of APP, compared to Akt inhibited cells, suggesting that BDNF acts through the Akt/GSK3 signalling axis to shift APP processing towards the non-amyloidogenic pathway (Baranowski et al., 2024). Ultimately, BDNF exerts pleiotropic effects that promote neuronal health and cognition; therefore, changes to BDNF concentration and expression, whether centrally or peripherally, may contribute to the onset or risk of AD.

## Skeletal muscle BDNF and AD

Recent research has highlighted the important role of skeletal muscle in dictating cognitive health, particularly showing the negative impact of sarcopenia on AD risk (Ceylan et al., 2023; Miyazaki et al., 2021). Declines in skeletal muscle size, strength and endurance – all characteristics of sarcopenia (Cruz-Jentoft et al., 2018; Filippin et al., 2015), can limit mobility and an individual's capacity to engage in regular exercise. Physical inactivity associated with sarcopenia can increase the risk of AD through various mechanisms, including weight gain, insulin resistance and impaired whole-body glucose regulation. Furthermore, regular exercise can increase the expression and protein content of various factors that can positively benefit cognitive health, including BDNF. Peripheral tissues are able to produce their own supply of BDNF, and skeletal muscle BDNF content has been shown to be lower in cases of muscle loss and obesity (Ceylan et al., 2023; Miyazaki et al., 2021). Though the exact role of BDNF in skeletal muscle is still up for debate, BDNF expression has been observed to increase in skeletal myofibres after repetitive muscle contractions (Matthews et al., 2009) and even during muscle repair (Liem et al., 2001). In fact, recent experiments with muscle-specific BDNF knockout (KO) mice have provided insight into the potential role of BDNF in skeletal muscle, including muscle repair, fat oxidation, mitochondrial quality and insulin sensitivity.

Skeletal muscle repair is an essential process for restoring optimal function after injury and disuse. However, as we age and in conditions such as sarcopenia, the ability of skeletal muscle to undergo repair and regeneration is impaired. This is largely due to a decline in the number and functionality of satellite cells, which ultimately contributes to the development of sarcopenia (Huo et al., 2022; Muñoz-Cánoves et al., 2020; Sousa-Victor et al., 2015). Recent research has implicated BDNF in muscle repair, where it can regulate myoblast fusion and satellite cell population (Liem et al., 2001; Mousavi & Jasmin, 2006). Specifically, Mousavi and Jasmin (2006) first discovered that BDNF was expressed in proliferating muscle satellite cells, suggesting that it might regulate satellite cell population. This was later confirmed with experiments in muscle-specific BDNF KO mice that exhibited sharp declines in satellite cell number, as well as myoblast proliferation and differentiation (Clow & Jasmin, 2010). In turn, primary myotubes derived from muscle-specific BDNF KO mice were smaller than those found in wild-type control; however, this effect could be reversed with exogenous BDNF treatment.

Furthermore, in response to cardiotoxin-induced injury, Clow and Jasmin (2010) found that muscle-specific BDNF KO mice (sex of the animals not reported) had delayed muscle regeneration with lowered presence of regenerative fibres that coincided with lowered expression of myogenic factors, including myogenin, myoD and embryonic myosin-heavy chain (Clow & Jasmin, 2010). Thus, BDNF plays a crucial role in regulating muscle repair and satellite cell function, and a decline in muscle BDNF with ageing may contribute to sarcopenia, which in turn could increase the risk of AD.

BDNF not only regulates muscle size and repair, but it may also have a positive impact on various aspects of skeletal muscle quality and function, especially at the level of the mitochondria. The mitochondria play a crucial role in regulating skeletal muscle endurance, insulin sensitivity, and providing energy for protein synthesis, all of which are compromised with sarcopenia (Bellanti et al., 2021; Coen et al., 2018), ultimately contributing to an increased risk of AD. In contrast with sarcopenia, regular exercise is known to stimulate mitochondrial biogenesis and respiration which enhances fatty acid oxidation and can help regulate levels of adiposity (Alizadeh Pahlavani et al., 2022; Joseph et al., 2016). After exercise, there is an increase in BDNF mRNA and protein expression in human skeletal muscle, and this increase in muscle BDNF has been shown to play a critical role in regulating mitochondrial fatty acid oxidation, mainly through the activation of adenosine monophosphate-activated protein kinase (AMPK) (Ahuja et al., 2022; Matthews et al., 2009). Additionally, AMPK plays a role in over-all mitochondrial health (Herzig & Shaw, 2018), and therefore BDNF-induced activation of AMPK may be a useful technique in targeting all mitochondria-related aspects of sarcopenia. Furthermore, findings from female muscle-specific BDNF KO mice demonstrate the vital role of BDNF in preserving mitochondrial quality and function by activating mitochondrial fission and mitophagy (Ahuja et al., 2022). Previous work concluded that BDNF is not secreted into circulation by skeletal muscle (Matthews et al., 2009); however, there is new evidence suggesting that muscle is capable of releasing BDNF into circulation. Both electrically stimulated C2C12 muscle cells and human myotubes were found to release BDNF into cell media (Fulgenzi et al., 2020). Further, muscle-specific BDNF KO mice had lower circulating levels of BDNF compared to a wild-type control, indicating that muscle does release BDNF into the periphery and thus can contribute to the total pool of circulating BDNF (Fulgenzi et al., 2020). In fact, BDNF secreted by differentiated human myoblasts was shown to act on human islet cells through its receptor, TrkB, to stimulate insulin secretion (Fulgenzi et al., 2020). Muscle-specific BDNF KO in female mice resulted in an accumulation of dysfunctional mitochondria, which impaired fatty acid oxidation in response to high-fat feeding, ultimately leading to greater weight gain and insulin resistance (Chan et al., 2015). Even under fasted conditions, female muscle-specific BDNF KO mice displayed insulin resistance and hyperglycemia along with muscle atrophy and weakness (Yang et al., 2019). Thus, BDNF in muscle not only affects the size and function of muscle in female mice, but it can also influence whole-body fat levels and insulin sensitivity. However, it will be important to examine sex differences that may arise with this response in future studies. Nevertheless, this demonstrates the significance of BDNF in the risk of sarcopenia and obesity and, ultimately, in the development of AD.

## BDNF and adipose tissue

With sarcopenic obesity there is an imbalance between fat mass and lean muscle mass, with a disproportionally high level of adipose tissue, particularly visceral adipose tissue (Perna et al., 2018). Excess adipose tissue with sarcopenic obesity contributes to a state of chronic low-grade inflammation and metabolic dysfunction (Makki et al., 2013). Though the accumulation of both subcutaneous and visceral adipose tissues is associated with sarcopenic obesity, visceral adipose tissue is more closely associated with inflammation and poorer metabolic profiles in sarcopenic obese patients (Perna et al., 2018). Whether or not visceral adipose tissue has a direct influence on AD development remains up for debate; however, Isaac et al. (Isaac et al., 2011) examined the relationship between abdominal adiposity and brain structure and function in healthy older adults and found that visceral adipose tissue had a negative effect on verbal memory, attention and significantly lowered hippocampal volumes. Additionally, Nyberg et al. (Nyberg et al., 2020) observed similar results in adults ranging from 20 to 80 years of age, in which body fat had negative implications on subcortical grey matter volume, hippocampal volume and memory. These results worsened with age and are further exacerbated by increased body fat (Nyberg et al., 2020). It is suspected that the role visceral adipose tissue plays in whole-body inflammation and metabolic dysfunction may lead to an increased risk of AD development. Specifically, inflammatory cytokines released from adipose tissue in both humans and high-fat diet-induced obesogenic male mice can result in neuro-inflammation and contribute to AD pathogenesis (Kim et al., 2022; Puig et al., 2012). Further, excess adiposity results in the ectopic accumulation in other tissues, such as skeletal muscle, thus contributing to a reduction in skeletal muscle insulin sensitivity and altered glucose homeostasis, further contributing to the amyloid load in the brain, promoting AD pathogenesis (Zhou et al., 2020).

The discovery of BDNF in adipose tissue was made by Hausman et al. (2006) during a porcine adipose tissue microarray study and was reported as an adipose tissue-secreted cytokine. A later study by Bernhard et al. (2013) examined BDNF expression in human adipose tissue, detecting its expression in both subcutaneous and visceral adipose tissue. Additionally, BDNF expression in subcutaneous adipose tissue had a negative correlation with obesogenic markers and adipocyte differentiation (Ikeda et al., 2018). Corroborating this finding, both BDNF heterozygous and adipose-specific BDNF KO mice had significantly increased body weights and adiposity compared to controls (Kernie et al., 2000; Nakagomi et al., 2015); meanwhile, injections of 7,8-dihydroxyflavone (7,8-DHF), a BDNF mimetic, reduce body weight and adiposity in diet-induced obese mice (Chan et al., 2015).

In a study by Colitti and Montanari (2020), it was found that BDNF treatment in 3T3-L1 cells enhanced mitochondrial morphology and white adipose tissue browning as dynamin-related protein-1 and uncoupling protein-1-protein expression increased after 24 h of BDNF exposure. This suggests that BDNF may be a thermogenic stimulator in adipose tissue, potentially promoting adipocyte browning. Adipose tissue browning has been suggested as an attractive therapeutic target for metabolic diseases, such as sarcopenic obesity, as it improves whole body metabolism, insulin sensitivity and glucose uptake. As AD is considered a metabolic disease, adipose tissue browning and the subsequent improvement in metabolism may be a potential mechanism for the prolongment of AD onset (Tayanloo-Beik et al., 2023). The exact role BDNF plays in the browning of adipose tissue is still novel and requires further research; regardless, BDNF injections reduce body weight and adiposity in obese mice (Chan et al., 2015), which reduces inflammation and improves insulin sensitivity, highlighting their potential therapeutic use in whole-body metabolic disorders.

## Sex differences, AD and sarcopenic obesity

An important consideration when discussing AD and its comorbidities is biological sex differences. The menopausal transition is considered a great risk factor for the onset of AD, and AD diagnoses are typically higher in women than men (Beam et al., 2018; Mosconi et al., 2018). Additionally, post-menopausal women are at a greater risk of sarcopenia as the loss of oestrogens accelerates the loss of muscle mass and strength, which further contributes to an increased risk of obesity as the reductions in oestradiol lead to a redistribution of adipose tissue towards visceral or abdominal depots (Kodoth et al., 2022). Oestrogens are considered anti-inflammatory by combating increases in inflammatory cytokine activity. However, reductions in oestrogens throughout menopause cause an imbalance of pro- and anti-inflammatory pathways, potentially exacerbating metabolic dysfunction, ultimately increasing the risk of comorbidities (Martin-Millan & Castaneda, 2013; Stefanska et al., 2015).

The onset of sarcopenic obesity between sexes differs: women experience greater onset between the ages of 60 and 79, whereas men experience greater onset over the age of 80 (Bouchard et al., 2009; Hwang & Park, 2023). Alterations to sex hormones may be a contributing factor, and BDNF may also play a role. The *bdnf* gene contains an estrogen response element (ERE), that when activated can increase the transcription of BDNF (Berchtold et al., 2001; Sohrabji et al., 1995). Potentially due to this ERE, circulating BDNF concentrations tend to be higher in women compared to men (Wei et al., 2017). In ovariectomized (OVX) mice, a commonly used preclinical menopause model, BDNF expression is significantly lower 4 and 8 weeks post-surgery (Tao et al., 2020). In women, BDNF plasma concentrations fluctuate throughout the menstrual cycle where BDNF is higher in the luteal phase compared to the follicular phase (Begliuomini et al., 2007). Further, BDNF concentrations significantly decrease after the onset of menopause compared to premenopausal women in the luteal phase and postmenopausal women taking hormone replacement therapy (Begliuomini et al., 2007). BDNF perfusion in the brains of middle-aged OVX rats has been shown to rescue synaptic plasticity (Kramár et al., 2012). Further, 10 days of peripheral BDNF treatment in aged female mice rescued ovarian function by increasing ovarian weight, number of follicles and oocytes and blood oestrogen levels (Liu et al., 2023). Therefore, BDNF may prove to be a useful therapeutic strategy in postmenopausal women, potentially protecting them from both AD and sarcopenic obesity.

Much of the current research focuses on male subjects and their response to BDNF treatment. Though there are studies that utilize female subjects (Ahuja et al., 2022; Yang et al., 2019), there are few that highlight specific sex differences in response to BDNF in the context of both sarcopenic obesity and the brain. Therefore, it is unclear whether BDNF may have a stronger effect in one sex or the other, and as such, future research should look to include both sexes to tease out the potential differences.

## Conclusions

This review highlights AD as a metabolic disorder with sarcopenic obesity being a major contributor through the loss of muscle mass and strength paired with an excessive gain in adipose tissue. Recent research has shown that targeting BDNF, particularly in skeletal muscle and adipose tissue, can provide benefits to whole-body metabolism that may attenuate AD pathology. This is in

addition to the well-known benefits that BDNF exerts directly on the brain. Thus, BDNF may represent a viable target for AD that acts on several metabolic pathways across various tissues, ultimately converging on cognitive health.

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

## Additional information

### Competing interests

No competing interests declared.

## Author contributions

E.N.C.: Conception or design of the work; drafting the work or revising it critically for important intellectual content; final approval of the version to be published; agreement to be accountable for all aspects of the work. P.J.L.: Conception or design of the work; drafting the work or revising it critically for important intellectual content; final approval of the version to be published; agreement to be accountable for all aspects of the work. P.D.-G.: Conception or design of the work; drafting the work or revising it critically for important intellectual content; final approval of the version to be published; agreement to be accountable for all aspects of the work. V.A.F.: Conception or design of the work; drafting the work or revising it critically for important intellectual content; final approval of the version to be published; agreement to be accountable for all aspects of the work. R.E.K.M.: Conception or design of the work; drafting the work or revising it critically for important intellectual content; final approval of the version to be published; agreement to be accountable for all aspects of the work.

## Funding

Canadian Government | Natural Sciences and Engineering Research Council of Canada (NSERC): Rebecca E.K. MacPherson, RGPIN-2017-0 3904; Canadian Government | Canadian Institutes of Health Research (CIHR): Val A. Fajardo, 232 838.

## Keywords

adipose tissue, ageing, brain, crosstalk, skeletal muscle

## Supporting information

Additional supporting information can be found online in the Supporting Information section at the end of the HTML view of the article. Supporting information files available:

**Peer Review History**

