## [Peer Review History · The Journal of Physiology]

The link between sarcopenic obesity and Alzheimer's disease: a BDNF point of view

Emily N Copeland, Paul J LeBlanc, Paula Duarte-Guterman, Val Andrew Fajardo, and Rebecca EK MacPherson
DOI: 10.1113/JP288032

Corresponding author(s): Rebecca MacPherson (rmacpherson@brocku.ca)

The following individual(s) involved in review of this submission have agreed to reveal their identity: Gilmara Gomes Assis (Referee #2)

Review Timeline:	Submission Date:	31-Oct-2024
	Editorial Decision:	16-Dec-2024
	Revision Received:	11-Jan-2025
	Accepted:	16-Jan-2025

Senior Editor: Laura Bennet

Reviewing Editor: Mathew Piasecki

Transaction Report:

Dear Professor MacPherson,

Re: JP-TR-2024-288032 "The link between sarcopenic obesity and Alzheimer's disease: a BDNF point of view" by Emily N Copeland, Paul J LeBlanc, Paula Duarte-Guterman, Val Andrew Fajardo, and Rebecca EK MacPherson

Thank you for submitting your manuscript to The Journal of Physiology. It has been assessed by a Reviewing Editor and by 2 expert referees and we are pleased to tell you that it is acceptable for publication following satisfactory revision.

ABSTRACT FIGURES: Authors may use The Journal's premium BioRender account to create/redraw their Abstract Figures (and any other suitable schematic figure). Information on how to access this account is here: <https://physoc.onlinelibrary.wiley.com/journal/14697793/biorender-access>.

REVISION CHECKLIST: Upload a full Response to Referees file. To create your 'Response to Referees' copy all the reports, including any comments from the Senior and Reviewing Editors, into a Microsoft Word, or similar, file and respond to each point, using font or background colour to distinguish comments and responses and upload as the required file type.

We look forward to receiving your revised submission.

Yours sincerely,

Laura Bennet
Senior Editor

EDITOR COMMENTS

Reviewing Editor:

Your paper has been reviewed by 2 experts in this field and although the collective opinion is positive, both reviewers recommend a number of amendments. Please pay particular attention to the accurate definition of BDNF, or provide an argument for its classification as a hormone if this is disputed.

Please also see 'Required Items' below.

REFEREE COMMENTS

Referee #1:

In this review, Copeland et al. describe the role of Brain-derived neurotrophic factor (BDNF) in Alzheimer's disease (AD) and sarcopenic obesity, suggesting that BDNF may play a critical role in multiple age-related diseases. Current literature is well summarized. This review highlights the need for future investigation. I have minor comments only.

1) Line 115: "BDNF is a neurotrophic hormone". BDNF is not an hormone.

2) The authors discussed the link between BDNF and AD using studies in cell cultures and animal models. They may want to cite studies in humans as well (serum BDNF in AD patients, BDNF expression in postmortem brains of AD patients, etc...)

3) The graphical abstract could be improved by clearly highlighting AD versus non-AD condition.

Referee #2:

The manuscript "The link between sarcopenic obesity and Alzheimer's disease: a

BDNF point of view" is an extended, well-anchored literature review that presents a synthesis of the mechanisms related to BDNF that are involved in sarcopenic obesity, precisely obesity, and AD.

The qualitative synthesis is good and the text is well written. I have only a small criticism that I would like for the authors to reflect about.

BDNF is a dimeric protein mainly synthesized by neural tissue, but also in muscle, endothelial, adipose and other tissues. It is expected a local role for BDNF (autocrine and paracrine), regarding that it signals through receptors tyrosine kinase, which work as signal amplifiers and evoke many pathways related to cell growth, differentiation, and survival, and exhibit an oncogenic potential.

The expression of BDNF across tissues other than neural is rather small (DOI: 10.3389/fnmol.2021.638176), and, as reported by Matthews et al., 2009 which applied an exercise model to detect that the expression of BDNF is increased in response to muscle contraction, there no support for a systemic (hormonal) role for BDNF via circulation. Instead, blood BDNF concentrations likely derive from blood cells production (DOI: 10.1249/MSS.0b013e31825ab69b) or, perhaps, a leakage from the central nervous system (DOI: 10.1113/expphysiol.2009.048512).

In sum, the studies support the involvement of oxidative metabolism in the regulation of BDNF in various peripheral tissues (DOI: 10.26402/JPP.2018.3.12), but the classification as 'protein hormone' (line 115) shall be carefully revised, in my opinion.

minor revision:

-line 115 protein hormone

REQUIRED ITEMS

- Please include an Abstract Figure file, as well as the Figure Legend text within the main article file (currently the legend is missing). The Abstract Figure is a piece of artwork designed to give readers an immediate understanding of the Review Article and should summarise the main conclusions. If possible, the image should be easily 'readable' from left to right or top to bottom. It should show the physiological relevance of the Review so readers can assess the importance and content of the article. Abstract Figures should not merely recapitulate other figures in the Review. Please try to keep the diagram as simple as possible and without superfluous information that may distract from the main conclusion of the Review. Abstract Figures must be provided by authors no later than the revised manuscript stage and should be uploaded as a separate file during online submission labelled as File Type 'Abstract Figure'. Please ensure that you include the figure legend in the main article file. All Abstract Figures will be sent to a professional illustrator for redrawing and you may be asked to approve the redrawn figure before your paper is accepted.

- Author profile(s) must be uploaded via the submission form. Authors should submit a short biography (no more than 100 words for one author or 150 words in total for two authors) and a portrait photograph of the two leading authors on the paper. These should be uploaded and clearly labelled together in a Word document with the revised version of the manuscript. Any standard image format for the photograph is acceptable, but the resolution should be at least 300 DPI and preferably more. A group photograph of all authors is also acceptable, providing the biography for the whole group does not exceed 150 words.

END OF COMMENTS

The manuscript "The link between sarcopenic obesity and Alzheimer's disease: a BDNF point of view" is an extended, well-anchored literature review that presents a synthesis of the mechanisms related to BDNF that are involved in sarcopenic obesity, precisely obesity, and AD.

The qualitative synthesis is good and the text is well written.

English language seems fine in my judgement.

I have only a small criticism that I would like for the authors to reflect about.

BDNF is a dimeric protein mainly synthesized by neural tissue, but also in muscle, endothelial, adipose and other tissues. It is expected a local role for BDNF (autocrine and paracrine), regarding that it signals through receptors tyrosine kinase, which work as signal amplifiers and evoke many pathways related to cell growth, differentiation and survival, and exhibit an oncogenic potential.

The expression of BDNF across tissues other than neural is rather really small (DOI: 10.3389/fnmol.2021.638176), and, as reported by Matthews et al., 2009 which applied an exercise model to detect that the expression of BDNF is increased in response to muscle contraction, there no support for a systemic (hormonal) role for BDNF via circulation. Instead, blood BDNF concentrations likely derive from blood cells production (DOI: 10.1249/MSS.0b013e31825ab69b) or, perhaps, a leakage from the central nervous system (DOI: 10.1113/expphysiol.2009.048512).

In sum, the studies support an involvement of the oxidative metabolism in the regulation of BDNF in various peripheral tissues (DOI: 10.26402/JPP.2018.3.12), but the classification as 'protein hormone' (line 115) shall be carefully revised, in my opinion.

Response Letter for JP-TR-2024-288032

General Response to Reviewers:

We would like to thank the reviewers for thorough review of our manuscript and for their constructive and helpful comments. We have responded to each of the points carefully. For clarity, the reviewers' comments have been italicized and our responses are in bold font. In consideration with the following responses to the reviewer's concerns and corresponding changes to our manuscript, we are hopeful that our manuscript will be suitable for publication.

EDITOR COMMENTS

Reviewing Editor:

Your paper has been reviewed by 2 experts in this field and although the collective opinion is positive, both reviewers recommend a number of amendments. Please pay particular attention to the accurate definition of BDNF, or provide an argument for its classification as a hormone if this is disputed.

Please also see 'Required Items' below.

REFEREE COMMENTS

Referee #1:

In this review, Copeland et al. describe the role of Brain-derived neurotrophic factor (BDNF) in Alzheimer's disease (AD) and sarcopenic obesity, suggesting that BDNF may play a critical role in multiple age-related diseases. Current literature is well summarized. This review highlights the need for future investigation. I have minor comments only.

1) Line 115: "BDNF is a neurotrophic hormone". BDNF is not an hormone.

We thank the reviewer for their comment. This was an uncaught error and has been changed to neurotrophic factor (line 114).

2) The authors discussed the link between BDNF and AD using studies in cell cultures and animal models. They may want to cite studies in humans as well (serum BDNF in AD patients, BDNF expression in postmortem brains of AD patients, etc...)

We thank the reviewer for this suggestion and have included human serum analysis data in lines 146-149. Additionally, we have included a study by Fulgenzi et al., that uses both murine models and human derived cells to examine the role of BDNF and muscle (found on lines 231-242).

3) The graphical abstract could be improved by clearly highlighting AD versus non-AD condition.

We thank the reviewer for their comment and have reconfigured the graphical abstract to improve clarity.

Referee #2:

The manuscript "The link between sarcopenic obesity and Alzheimer's disease: a BDNF point of view" is an extended, well-anchored literature review that presents a synthesis of the mechanisms related to BDNF that are involved in sarcopenic obesity, precisely obesity, and AD.

The qualitative synthesis is good and the text is well written. I have only a small criticism that I would like for the authors to reflect about.

BDNF is a dimeric protein mainly synthesized by neural tissue, but also in muscle, endothelial, adipose and other tissues. It is expected a local role for BDNF (autocrine and paracrine), regarding that it signals through receptors tyrosine kinase, which work as signal amplifiers and evoke many pathways related to cell growth, differentiation, and survival, and exhibit an oncogenic potential.

The expression of BDNF across tissues other than neural is rather small (DOI: 10.3389/fnmol.2021.638176), and, as reported by Matthews et al., 2009 which applied an exercise model to detect that the expression of BDNF is increased in response to muscle contraction, there no support for a systemic (hormonal) role for BDNF via circulation. Instead, blood BDNF concentrations likely derive from blood cells production (DOI: 10.1249/MSS.0b013e31825ab69b) or, perhaps, a leakage from the central nervous system (DOI: 10.1113/expphysiol.2009.048512).

In sum, the studies support the involvement of oxidative metabolism in the regulation of BDNF in various peripheral tissues (DOI: 10.26402/JPP.2018.3.12), but the classification as 'protein hormone' (line 115) shall be carefully revised, in my opinion.

minor revision:

-line 115 protein hormone

We thank the reviewer for this comment and acknowledge that this was an error in wording. We would like to note, however, that although Matthews et al., concluded that BDNF is not secreted by muscle, recent work by Fulgenzi et al., (2020) found that upon being electrically stimulated, C2C12 cells and human-derived myotubes both released BDNF into the cell media. Additionally, muscle-specific BDNF KO mice were found to have lower circulating levels of BDNF, suggesting that muscle-induced release of BDNF contributes to the amount of

circulating BDNF. Therefore, the secretion of BDNF from muscle likely contributes to its systemic roles, though not as a hormone *per se*. This information has been included in the manuscript and can be found in lines 231-242.

Fulgenzi, G., et al. (2020). "Novel metabolic role for BDNF in pancreatic beta-cell insulin secretion." Nat Commun 11(1): 1950.

Dear Professor MacPherson,

Re: JP-TR-2025-288032R1 "The link between sarcopenic obesity and Alzheimer's disease: a BDNF point of view" by Emily N Copeland, Paul J LeBlanc, Paula Duarte-Guterman, Val Andrew Fajardo, and Rebecca EK MacPherson

We are pleased to tell you that your paper has been accepted for publication in The Journal of Physiology.

Authors should note that it is too late at this point to offer corrections prior to proofing. Major corrections at proof stage, such as changes to figures, will be referred to the Editors for approval before they can be incorporated. Only minor changes, such as to style and consistency, should be made at proof stage. Changes that need to be made after proof stage will usually require a formal correction notice.

Yours sincerely,

Laura Bennet
Senior Editor
The Journal of Physiology

P.S. - You can help your research get the attention it deserves! Check out Wiley's free Promotion Guide for best-practice recommendations for promoting your work at www.wileyauthors.com/eoo/guide. You can learn more about Wiley Editing Services which offers professional video, design, and writing services to create shareable video abstracts, infographics, conference posters, lay summaries, and research news stories for your research at www.wileyauthors.com/eoo/promotion.

IMPORTANT NOTICE ABOUT OPEN ACCESS: To assist authors whose funding agencies mandate public access to published research findings sooner than 12 months after publication, The Journal of Physiology allows authors to pay an Open Access (OA) fee to have their papers made freely available immediately on publication.

You can check if your funder or institution has a Wiley Open Access Account here: <https://authorservices.wiley.com/author-resources/Journal-Authors/licensing-and-open-access/open-access/author-compliance-tool.html>.

EDITOR COMMENTS

Reviewing Editor:

Thanks you for responding to all comments.

The following administrative items need to be added to your article at proof stage:

- Your MS must include a complete "Additional information section" with the following 4 headings and content:

Competing Interests: A statement regarding competing interests. If there are no competing interests, a statement to this effect must be included. All authors should disclose any conflict of interest in accordance with journal policy.

Author contributions: Each author should take responsibility for a particular section of the study and have contributed to writing the paper. Acquisition of funding, administrative support or the collection of data alone does not justify authorship;

these contributions to the study should be listed in the Acknowledgements. Additional information such as 'X and Y have contributed equally to this work' may be added as a footnote on the title page.

It must be stated that all authors approved the final version of the manuscript and that all persons designated as authors qualify for authorship, and all those who qualify for authorship are listed.

Funding: Authors must indicate all sources of funding, including grant numbers. If authors have not received funding, this must be stated.

It is the responsibility of authors funded by RCUK to adhere to their policy regarding funding sources and underlying research material. The policy requires funding information to be included within the acknowledgement section of a paper. Guidance on how to acknowledge funding information is provided by the Research Information Network. The policy also requires all research papers, if applicable, to include a statement on how any underlying research materials, such as data, samples or models, can be accessed. However, the policy does not require that the data must be made open. If there are considered to be good or compelling reasons to protect access to the data, for example commercial confidentiality or legitimate sensitivities around data derived from potentially identifiable human participants, these should be included in the statement.

Acknowledgements: Acknowledgements should be the minimum consistent with courtesy. The wording of acknowledgements of scientific assistance or advice must have been seen and approved by the persons concerned. This section should not include details of funding.

REFEREE COMMENTS

Referee #2:

The revisions were well provided.